# Understanding the Wicked Problem of Providing Accessible Housing for the Ageing Population in Sweden

**DOI:** 10.3390/ijerph18031169

**Published:** 2021-01-28

**Authors:** Oskar Jonsson, Joakim Frögren, Maria Haak, Björn Slaug, Susanne Iwarsson

**Affiliations:** 1Department of Health Sciences, Lund University, SE-221 00 Lund, Sweden; joakim.frogren@med.lu.se (J.F.); maria.haak@hkr.se (M.H.); bjorn.slaug@med.lu.se (B.S.); susanne.iwarsson@med.lu.se (S.I.); 2Department of Nursing and Integrated Health Sciences, Kristianstad University, SE-291 88 Kristianstad, Sweden

**Keywords:** ageing in place, built environment, decision support, housing accessibility, housing and health, housing provision, research circle, trade-offs, planning, public health

## Abstract

While accessible housing is known as important to promote healthy ageing, the societal issue of providing accessible housing for the ageing population bears the characteristics of a “wicked problem”. The aim of this study was to gain a better understanding of crucial variables for decision-making about the provision of accessible housing for the ageing population in Sweden. Materials used for a deductive content analysis were elicited through a research circle involving three researchers and twelve non-academic representatives. Brown and colleagues’ conceptual five-dimension framework to address wicked problems was used for the understanding of crucial variables in decision-making about housing provision. The findings show that such reasoning is dominated by the socioeconomic dimension. Findings in the biophysical dimension reveal well-known challenges pertaining to the definition and interpretation of the concept of accessibility and its operationalization. The dimensions are intertwined in a complex manner, which is essential for effective and efficient decision-making. The findings could make decision-makers aware of the diversity of individual thinking involved when addressing this wicked problem. Acting upon the crucial variables identified in this study could contribute to progressive decision-making and more efficient ways to develop and provide accessible housing to promote health ageing.

## 1. Introduction

In their Housing and Health Guidelines report, the World Health Organization (WHO) [1] declared that “Poor housing conditions are one of the mechanisms through which social and environmental inequality translates into health inequality, which further affects the quality of life and wellbeing” [1] (p. xv). People with the lowest incomes, minority groups, indigenous people, woman, single parent families, and people ageing with functional limitations are overrepresented as residents in unsuitable housing, and thus prone to inequality and inequity [1]. Still, many questions remain about multiple interrelations and how the need for housing that creates conditions for good health for as many people as possible can be met [2].

Ageing societies are facing major housing challenges. There is an urgent need to better understand and provide for the diversity of housing requirements of senior citizens. Key issues are accessibility, adaptability, affordability, attractiveness, availability of services, facilities and infrastructure, habitability, location, and security for tenure. With this study, we have the ambition to contribute to developments responding to the needs of accessible housing for the ageing population.

Attempts to create comprehensive conceptual frameworks of housing and health have been made, but because the phenomena involved are multidimensional and changing over time, such attempts have also been criticized. One recommendation is to direct the focus of interest to a specific aspect of housing and find fruitful concepts and theories from that point of view [3]. Accordingly, in this study, we focused on housing accessibility as a prerequisite for healthy ageing.

## 2. Housing and Health

Globally, numerous research outputs on housing and health have recently been achieved [1,4,5]. A shared objective is a better understanding of various environmental variables, their interrelations and how they positively or negatively affect health and wellbeing. To bridge research with policy and practice, WHO [1] provided evidence-based recommendations to improve housing conditions and reduce the health burden. However, prioritizations and implementations vary and depend on national and contextual factors. Moreover, political will and muscle, as well as cross-sectorial collaboration, are required to achieve tangible progress.

### 2.1. Housing Accessibility

Accessibility is a basic quality of housing, which is not just about physical features in the environment but defined as the encounter between a person’s or group’s functional capacity and the demands in the physical environment [1,6]. A physical environment that exposes people to risk of injuries, isolation, and stress, hampers daily activities and participation in social life has negative health effects [7], and increase the burden on health care and social services [1]. With population ageing an increasing proportion of older adults will live longer with functional limitations [1]. Accordingly, the prevalence and magnitude of accessibility problems are likely to increase unless large-scale actions to improve accessibility are taken to resolve issues in the physical housing environment. The prevailing ageing-in-place policy—a priority for many senior citizens and a policy objective particularly strong in the Scandinavian countries [8]—further increases the importance of providing accessible housing. Ageing in place includes enabling people to live later life in ordinary housing as long as possible.

#### Housing Accessibility in the Swedish Context

In Sweden, despite high housing standards in the ordinary housing stock and municipal obligations to provide individual housing adaptation grants, there is a high prevalence of physical environmental barriers and considerable accessibility problems for senior citizens with functional limitations [7,9]. 

From an institutional and political point of view, the Swedish government’s general housing policy implies that the municipalities are responsible for providing adequate housing for their inhabitants within the ordinary housing stock. They are obliged to develop and ratify “housing supply action plans” [10], including objectives and planned efforts to achieve the objectives set for existing as well as new housing. These plans should be based on relevant national and regional goals, plans and programs, and be developed in collaboration with the municipal building administration, the municipal health care and social services administration, public, and private housing companies, and relevant communities of interest (non-governmental organizations, NGO). To support actions to increase housing accessibility, financial grants from the government can be awarded to construction companies, housing companies, and private property owners to compensate for their investments [11]. In addition, there are grants for financial support for individual housing adaptations provided by the municipalities [12]. 

In other words, planning and provisions for accessibility involve a variety of institutions and actors whose assessments and decisions affect one another in a tangled manner. Poor planning and administrative errors may cause gaps in the implementation of accessible housing and may result in negative health effects [13]. While accessibility is a problem that has been discussed over decades, the understanding of crucial variables for decision-making about accessible housing for the ageing population is insufficient.

## 3. A Conceptual Framework to Address Wicked Problems

To unwind, understand and communicate the multiple and complex relations between housing and health in the context of housing provision for the ageing population, innovative and shared conceptual frameworks that apply systems thinking are needed [5]. One way to tackle the uncertainty associated with complex planning problems is to open up for contributions from a diversity of academic disciplines and non-academic representatives to generate knowledge and develop ”open transdisciplinary modes of inquiry capable of meeting the needs of the individual, the community, the specialist traditions, and influential organizations” [14] (p. 4).

### 3.1. Wicked Problems

Many of the societal-environmental planning problems and challenges faced today are characterized by complex relationships, including a web of variables that directly or indirectly affect each other in an intricate way. About 50 years ago, Rittel & Webber [15] suggested referring to such issues, including public policies for housing and population health, as “wicked problems” and contrasted these to “tame problems”, that is, problems that can be treated as scientific problems possible to solve within the existing modes of inquiry and decision-making. What further distinguishes a “wicked problem” is that in order to give it justice, it was not considered possible to look at or assess it solely from one angle or approach. Instead, due to its complex nature, the problem required analysis based on values linked to a variety of actors. A “wicked problem” had no final solution but only the best possible solution at any particular time [15]. 

Using the notion of “wicked problems” we developed a point of departure for the present study, outlined in Table 1. Based on current literature [4,5,16], public policy [1,17], public debate, and experiences from our research, we related the definition of “wicked problems” to the provision of accessible housing.

Exemplified by Table 1, there is a high likelihood of conflicts between different views and interests [18] because a wicked problem requires an understanding of a variety of perspectives. These views and interests may be based on the individual, but they may also reflect the organizational structure of institutions or division within existing society. However, the pursuit of coherence in tackling a wicked problem is based on individual efforts and include multiple dimensions of individual thinking [19], although some individuals in a group setting usually have a greater ability than others to embrace complex problems. Based on this insight, Brown, and colleagues [19] proposed a conceptual framework with five dimensions of individual thinking to address wicked problems: biophysical, socioeconomic, ethical, aesthetic and sympathetic (Table 2). Wicked problems require consideration from all these dimensions and the mutual interactions between them, in order for their complexity to be fully recognized and truly understood. 

The biophysical dimension consists of what is created through our senses (including tools that extend our senses) and is objectively measurable in the biological and physical environment, which constitutes the scene of the problem at target. Applied to the wicked problem under study, research confirms that the material and spatial constituents of housing units are important for senior citizens, and their preferences and requirements need to be understood [4]. The socioeconomic dimension does not only include the social and economic reality of people but takes on a wider meaning. It is about “stories” reflecting how society was established, and existence in the form of infrastructure, objectives, and ways of life. This dimension encompasses economic and legal systems, and the importance people ascribe to the market and law, as well as the infrastructure set up to regulate these. It also encompasses the social, ethnical, cultural, and religious conditional approaches and living rules established and maintained by parts of or the entire population. The ethical dimension constitutes the ethical principles of what is considered morally right and wrong from an individual’s stance in relation to fellow human beings, other creatures, and the environment. These questions require a highly individual stance based on a moral or altruistic compass about what is right or wrong. While ethical questions might have ”answers” in existing policies and regulations, the answers are value-based and are thus changeable based on the values that prevail in society. However, each individual has the opportunity to take a personal stance on these issues and can accommodate change if existing policies and regulations go against one’s own stance. The aesthetic dimension is based on an objective, as well as subjective view of what is attractive on the basis of our sensitivity to the patterns in natural and social systems. The value of aesthetics and unobtrusiveness as well as the importance of avoiding an institutional look and unwanted associations with loss of independence is an example of a recurring attitudinal theme in relation to housing accessibility [20]. Research confirms that the aesthetic preferences of residents are not necessarily the same as those of architects and designers [21]. The sympathetic dimension refers to our ability to emotionally relate to, recognize or identify with what other people feel. While Brown and colleagues [19] termed this dimension “sympathetic”, but we consider that empathetic is a more appropriate term. This is because sympathy only includes the negative emotion register and distancing such as feeling compassion, grief, or concern for someone [22]. Empathy, on the other hand, involves both the ability to understand another person’s situation (cognitive empathy) and react emotionally on the basis of this (emotional empathy), and includes a broader emotional register and a closer distance [23]. We made use of the five dimensions proposed by Brown and colleagues [19] as a conceptual framework to identify pertinent variables in the context of the provision of accessible housing for senior citizens in Sweden.

### 3.2. Aim of the Study

Using wicked problems and the five dimensions as the conceptual framework underpinning the present study, the aim was to further the understanding of crucial variables for decision-making about provision of accessible housing for the ageing population in Sweden. 

## 4. Materials and Methods

### 4.1. Study Context

The study was conducted based on empirical materials originating from the initial phase of the project Decision Support System for Improved Accessibility in Multi-Family Housing [24]. The overarching aim of that project was to develop, test, and evaluate a new decision support system for improved accessibility in multi-family housing, and to contribute to efficient collaboration among professionals involved in health care, planning, and housing provision, as well as citizens. The project team consisted of researchers and two non-academic partners: a public housing company and a micro-sized software development company.

As part of laying the groundwork and creating conditions for the design of the new system, the aim of the initial phase of the larger project was to involve representatives for future potential users to gain an understanding of the potentials and challenges they envisaged with regard to the development, implementation, and commercialization of the new system. We used the research circle (RC) methodology as a joint educational and exploratory group setting to promote active collaboration between non-academic representatives and researchers [25]. The RC methodology encourages and provides opportunities for sharing knowledge, know-how and ways of knowing for mutual learning in the form of group dialogues on equal terms [26,27]. Findings from the RC relating to the specifications of the decision support system are presented elsewhere [28].

### 4.2. Participants and Recruitment

The RC participants consisted of twelve persons representing various categories of non-academic actors and institutions (i.e., key actors) and three researchers with different academic backgrounds (design sciences, health sciences and cognitive science) (Table 3). The key actors’ perspectives were considered as essential to better understand the complexities in individual thinking related to the wicked problem of providing accessible housing for the ageing population. The recruitment was based on the fact that the key actors were interested in and potential beneficiaries of the new knowledge produced from research on housing accessibility, had competence, knowledge, experience of and opinions on issues related to housing and health. With the ambition to create a dialogue on equal terms, the researchers and the key actors worked together as participants in the RC. Like the key actors, the researchers had and shared knowledge and opinions about the issue under study, although they assumed facilitative and probing roles.

We used purposive sampling [29] via two channels to recruit key actors: (1) A list of persons who had previously shown interest to participate in research at our research centre; (2) Persons who were encountered during the planning and start-up of the research project. Homogeneity was reached as the key actors were selected from among non-academic actors and institutions involved in housing and health issues. Heterogeneity was attained using professional position, organization, and sex as selection criteria. The key actors selected had a potential to be information-rich and were experts on organizational opportunities and constraints among different non-academic actors and institutions. An exclusion criterion was key actors not able to participate in all three RC sessions. In total, 25 persons were invited to participate whereof 12 accepted and participated together with the three researchers (Table 3). Representatives from one industry organization and four private housing companies were among those unable or not willing to participate.

### 4.3. Procedure

Three RC sessions were held, scheduled for three hours each and included audio-recorded presentations and discussions. Two of the authors (O.J. & M.H.) assumed the role of moderators. One of the authors (J.F.) participated and took notes in the third session. In the first session, the theme concerned how problems addressed by the new system were currently solved. The theme of the second session concerned the potential outcomes and applications of the new system. The theme of the third was about identifying potential customer segments and developing a draft business plan for the new system. The researchers hosted the first and third sessions that were held in conference rooms at the university. By a joint decision, a public housing company hosted the second session.

### 4.4. Ethics

According to current Swedish legislation, formal ethical approval is not required for studies that do not elicit material concerning sensitive information and do not include any intervention to humans. Following recommendations for proper research conduct [31], the key actors received written and verbal information about the study and had opportunities to ask questions. The key actors signed an informed consent at the start of the first session including information on voluntariness, the option to drop out, confidentiality, and the use of audio recordings.

### 4.5. Analysis Procedure

The empirical material consisted of the audio recordings from the RC sessions and the corresponding transcripts. One author (J.F.) transcribed the audio recordings according to Linell’s transcription level 2 [32] using NVivo [33], read the material several times while taking notes and summarizing thoughts in relation to the study aim. We applied deductive manifest content analysis according to Elo & Kyngäs [34], focusing on the ten properties of wicked problems [15] (see Table 1). We used Brown and colleagues’ [19] framework as a grid to sort critical variables involved in decision-making processes. Two authors (J.F. & O.J.) had iterative discussions throughout the analysis process. The data were reviewed for critical variables, coded in emerging categories and sorted into the five dimensions. The emerging findings were validated repeatedly through communication including all authors, as well as through input from interdisciplinary research seminars. A professional translator translated selected quotations to English, used to illustrate and contextualize the findings. In a last round of optimization, two of the authors finalized the findings (O.J. & S.I.).

## 5. Findings

All five dimensions of the framework were represented in the material. For the biophysical, socioeconomic and ethical dimensions several categories were identified (Table 4).

### 5.1. The Biophysical Dimension

#### 5.1.1. Different Opinions on the Meaning and Definition of Housing Accessibility Prevail

The individual interpretations of the concept of housing accessibility generated a diversified discussion, highlighting some differences between the categories of participants involved. A shared opinion was that the building regulations indicating the minimum acceptable level of physical properties in the built environment served as a foundation. However, an attitude that prevails in society is that housing accessibility only concerns people with physical limitations rather than being a right for all. The researchers argued for a relational definition between functional capacity of the individual(s) and barriers in the physical environment as a way to embrace the complexity of functional limitations, enabling cross-sectoral collaboration and various analyses and evaluations. The key actors agreed, but some of them highlighted the importance of including cognitive limitations (senior citizens’ organization representative) and barriers for social interactions (representatives from the municipal health care administrations and the national public authority). Moreover, there were different opinions about the level of detail necessary to understand accessibility problems, and the boundary between the dwelling, its immediate surroundings, the neighborhood, and the connecting local environment. For example, the senior citizens’ organization representative promoted a more comprehensive definition.
*… a problem is also that they are not really aware of all the shortcomings when you get older? If we now talk to... I mean the housing adaptation grants go to 70% to households where you are 65 years or older, so if we talk about issues regarding older people then it is, dementia, the dawning dementia is a huge problem, and yet housing companies continue when they repair, put in stoves where you have to press red buttons... which you do not understand as cognitively impaired*.(Senior citizens’ organization representative).
*So the solutions are not really adapted to*...(Business developer).
Not always...(Senior citizens’ organization representative).

The participants put forth that a common definition of housing accessibility is a critical, high-priority aspect to create cross-sectorial solutions, enabling the provision of accessible housing for the ageing population. 

#### 5.1.2. Systematic Inventories Are Warranted But Must be Comprehensive

The discussions revealed that a systematic inventory of environmental barriers in the existing housing stock is regarded as a critical variable. An important step is to establish databases and create conditions for scoring and weighting the biophysical dimension of housing accessibility against other measurements such as construction, energy consumption, and fire safety for prioritization and decision-making. According to the participants, such tools were currently lacking. Applying the definition that accessibility is determined by the concept of person-environment fit, the possibility to enter data on human functional capacity into databases with information about environmental barriers to support decision-making was much appreciated by the key actors. The participants emphasized that human physiology entails a variation in cognitive abilities. Dealing with variations in human functioning in a valid manner, combining data on the environmental and personal components of housing accessibility could produce useful information and enable classification of the degree of accessibility on the local, regional, and national levels, as well as on individual and group levels. 

The participants requested a reliable database that would make it possible to extract objective and comparable information. This was seen as potentially useful as a support for a diversity of non-academic actors and institutions and different kinds of decision-making. Examples given were support for housing companies when making decisions about possible retrofit actions to improve accessibility, such as zero-step entrances. Another was as support for people to find a new dwelling with the best possible accessibility based on their current or future functional limitations:
*Information [about housing accessibility] is of course important, both for those who want somewhere to live and those who want to build or own housing, as well as others such as the municipality who have to make repairs and compensate for faults and deficiencies*.(Municipal building administration).

#### 5.1.3. Evidence and Convincing Arguments for Housing Accessibility Are Important but Lacking

Lack of evidence on the effects of actions to improve accessibility for public health and societal economy was found to be a critical variable. Participants asked for evidence on how housing accessibility is related to health-related aspects such as injuries from falls, the need for home services and the opportunity to live a longer and healthier life. This kind of data was desirable to lay a foundation that could underpin cost-effective decisions made on regional and national levels, and to enable and motivate “matching” of housing based on individual functional ability. More research was asked for on this issue, and researchers were attributed an important role. The importance of having access to a model or “argumentation collection” rooted in research through which one could work proactively and convince others to think and/or act in a certain direction was stressed.
*If I stand in front of my board of directors, for example, and then I say that: yes, now I have a policy program here that will involve higher costs in maintenance programs and new production, but this we are prepared to take, because... and can I then prove that here we get happier tenants, and they will stay longer with us, and then we save this much money. If we don’t do this, we will have an alternative cost... Therefore, it is better to work proactively for these things in some way... it’s this collection of arguments I think we need to have with us*.(Housing company).

### 5.2. The Socioeconomic Dimension

#### 5.2.1. The Ageing-in-Place Policy is Significant for Decision-Making

One prevailing narrative highlighted as significant for decision-making was the ageing-in-place policy. This was said to guide societal efforts and strongly influence decision-making linked to the provision of accessible housing for the ageing population. However, there was some uncertainty regarding the interpretation of this prevailing policy objective. On the one hand, it could be interpreted as that people’s current dwellings are the places for them to age in no matter the costs, with individual housing adaptations, home health care and social services provided to avoid relocation to residential care facilities.
*So that is sort of the question… where does the individual’s responsibility come in? Can I stay as long as I want* [in my home] *and still get housing adaptation? To what extent?*(Housing company).

On the other hand, the ageing-in-place policy could be interpreted as broader and more proactive. For example, providing housing on the ordinary housing market that creates not only accessible but also attractive and favorable surroundings with conditions for social interaction, mutual support, and autonomy without individual societal support.
Yes, but I also think that this ageing-in-place policy should be interpreted as… a matter of avoiding relocations to residential care facilities, and actually support ageing in the ordinary housing stock but ageing in the right place.(Researcher).
*That’s how I think too... that you want to prevent people from moving to residential care facilities, that’s what you want to prevent*.(Housing company).

#### 5.2.2. Organization and Distribution of Resources Suffer from “Silo thinking”

The prevailing organizational structure was found to be critical, characterized by a “silo” mentality hampering different types of collaboration. One facet of this was expressed in the form of difficulties related to cross-sectorial communication:
*That someone from the care sector can talk to someone from the housing sector… I think that’s often where deep gaps are... the fact that you communicate around your own concepts and terms, and it also becomes very exclusionary if people don’t understand, so I think that is very important*.(National public authority).

A critical variable related to the organizational structure was that relevant information and knowledge were often available and used on the local level but not sufficiently shared regionally, nationally and across sectors. As to new construction, participants argued that the complexity and existing organizational “silos” resulted in actors often missing essential aspects due to lack of knowledge, miscommunication and no actor taking full responsibility:
*One does this, another does that… and then all of a sudden it turns out as it does, and then maybe it’s necessary to correct problems that arise, which then is done and so on... I mean, I have tried to map out the processes, in other words I have tried to identify where all the errors occur and... it’s like one long obstacle course*.(Municipal building administration).

According to the participants, these “silos” have contributed to a distribution of resources and responsibilities that to some extent has impaired the opportunities to take action to improve accessibility. This was considered to partly be due to the state not providing public housing companies with sufficient resources with regard to the level of societal responsibility they were interested in and expected to assume:
*Then we have a problem… it is this classic… that the money is in different coffers so everything we have to do is not economically profitable* [from a business economics perspective], *but it is socio-economically very profitable*. (Housing company).

Another critical variable highlighted by the participants representing housing companies was the European competition law, prohibiting companies from conducting any bargaining with municipalities. Despite the housing companies’ interest, ambitions and responsibilities, this hindrance to get appropriate financial compensation prevented major retrofit actions.

Thus, key actors from public housing companies said that major retrofit actions, currently with low financial compensation, resulted in increased housing costs for the tenants; sometimes to the extent that they could not afford it. The participants had diverse opinions concerning the practical consequences of such efforts under the current distribution of responsibilities and resources. Some put hope in existing and new technology to achieve sustainable development.
*It’s great with side sliding doors, then you have an air lock... there is technology for everything*.(Municipal building administration).
*But air locks cost... then you need two* [sliding doors]*, but it is possible that it will be so, that there will be a directive that you must have sliding doors. Yes, okay, then we do not have two hundred thousand, then we have three hundred thousand who can’t afford to live in this country*.(Housing company).

Some gave examples of efforts that turned out to be counter-productive, such as new construction projects with high degree of accessibility, but where apartments remained vacant:
*We have built... with an increased standard... there are larger bathrooms... no one was interested… Not until they have moved in and found out—well it was rather good... there are no thresholds, and it is possible to turn around. But pay for it? No. But, it may come*.(Housing company).

#### 5.2.3. Varying Practices and Competing Priorities among the Actors

Reflecting on the interest in accessibility matters among the general public, certain accessibility features were said to be valued by almost everyone. According to key actors from public housing companies, few housing applicants give priority to accessibility matters unless they face problems themselves, and few accessibility issues are raised by their tenants.

A critical variable found was the need of a financial policy that does not impact on peoples’ choices, making it possible for individuals to make choices about where to live in later life without being pressured by unwanted personal economic consequences. Participants felt that in the prevailing socioeconomic systems there were understandable explanations about why personal finances were prioritized over housing accessibility. Still, stronger emphasis and more proactive thinking about housing accessibility among citizens as well as actors in the planning, housing, and care sectors were seen as important. From a citizens’ perspective, this was argued partly for reasons related to democracy and the maintenance of the welfare state:
*I do not know if this is exactly what… but we must get citizens to be more involved in the development of welfare*.(Municipal health care administration).

The participants argued that although the municipal building administration should be the institution where much of the accessibility work was based, the interest in these issues was often low there. Those who worked there were said to be primarily interested in new construction rather than existing buildings, which was assumed to be one plausible explanation. For the drive and commitment towards the provision of accessible housing, the participants suggested that the municipal health care administration should be actively involved. Furthermore, it was alleged that building inspectors and administrators in the municipality had limited knowledge about housing accessibility. In smaller towns/urban areas and rural municipalities it could be a single person who had such a focus, but if there was no person with such special competence the responsibility ended up with the building permit administration. Similarly, key actors from housing companies declared that they lacked knowledge and competence in this area. The importance of acknowledging the mindsets of leaders of organizations and the creation of sustainable competence and culture on all levels within an organization was emphasized:
*Within each main category there must be several stakeholders working on this issue; it is not enough that it’s only one individual, that’s very important*... (Housing company).

A partial explanation for low competence in this area was a combination of a limited interest from the management, and the fact that to obtain accessibility knowledge, people were referred to texts of general recommendations and standards that they did not have time to go through. Moreover, housing companies claimed that they had to rely on accessibility consultancy firms. Their services were regarded by some as expensive, potentially unreliable, and not in line with the aim of becoming a more self-sufficient, competent, and learning organization. However, the opinions about consulting services were divided. The representative from the architecture and engineering consultancy emphasized that they not only had know-how but also contributed with new approaches and creative solutions. 

#### 5.2.4. Absence of Clear Housing Accessibility Guidelines and Goals 

The absence of reasonable and clear guidelines and goals regarding what was considered as “good enough” housing accessibility, both in terms of degrees and coverage ratio, was a crucial question from the point of view of housing companies:
*I mean... it’s about some kind of cost-benefit analysis on the societal level* [as it exists in other areas]*... then you should be able to have it here too*...(Senior citizens’ organization representative).
*But, I think we know, everyone sitting around this table, that is, that the main argument from the property developer is: why do we have to make 100% of our apartments accessible. Why isn’t 25% enough? It is not as if everyone is disabled*…(Housing company).

Hence, the importance of balancing use value with monetary cost was emphasized. Calculation of costs for investments in accessibility and alternative costs for individual housing adaptations, home health care, and social services on the municipal level was encouraged:
*It is then, for example, important for the municipality, I mean, in a municipal management, to see the alternative cost between either investing in making their own housing stock better and more accessible, as opposed to all the help that is needed in relation to the demographic development... So there you can, for the municipality, based on your own housing stock, and stuff like that, you should be able to calculate* [the cost savings] *by adding examples and cost calculations to them.*(Municipal building administration).

### 5.3. The Ethical Dimension

#### 5.3.1. Balance between Individual Freedom of Choice and Societal Solidarity

One issue that emerged in the discussion of the ageing-in-place policy was which approach to endorse when persons in later life wish to remain in, or move to, a less accessible dwelling that comes with the risk of requiring costly housing adaptation grants for the municipality, rather than move to a more accessible alternative. Some participants argued that the current laws, allowing municipalities to reject applications for individual housing adaptation grants, limited such risks. Discussing the responsibility of the individual to apply an ethical perspective, participants emphasized that on the one hand. it is important to contribute to fair and reasonable use of resources, on the other hand, it is important to let people make their own housing decisions. However, it also became apparent that the frequently mentioned issue of senior citizens’ relocation from less accessible villas to more accessible apartments was not regarded merely as a question of individual freedom of choice or societal solidarity. It also emerged that in a situation of housing shortage and overcrowded households, it could be framed as taking responsibility for using resources and the existing housing stock efficiently:
*We ran a project where we said: Anyone over the age of 75 living alone in a villa has 10 priority choices for accessibility adapted… for apartments they could enter and leave without hindrance; and then families with children bought the villas and this allowed the municipality to grow in population numbers without having to build even a single new square meter. That was absolutely the best*.(Housing company).

#### 5.3.2. Ambiguous Social Responsibilities of Housing Companies

The amount of social responsibility that public housing companies should take was discussed. The participants representing such institutions reported that in past decades they had been forced to take on an increased social responsibility:
*We have to have some kind of commercial profitability in what we do. We are not social services. We are a business when all is said and done. If people think we are social services and that it is our responsibility to look after the vulnerable groups, they might have the wrong idea. Now I am speaking for myself, but in reality we are functioning as social services at the moment… and it is a good thing we are because there are a lot of holes in the net these days that people are falling straight through*…(Housing company).

In addition, the importance of getting private housing actors to assume a more active role to provide accessible housing for the ageing population was emphasized. The participants argued that private housing actors should show greater interest in work aimed at improving housing accessibility, at least if such efforts were to be associated with increased value.

#### 5.3.3. Balance between Housing Accessibility and Housing Affordability

Another ethical issue related to the assumption that within the current socioeconomic systems, apartments in the older and often less accessible housing stock were said to allow some groups of individuals to have a home at all:
*There is a market for these cheap apartments as well*.(Housing company).
*There are a great many who need a simple apartment due to problems renting, and we don’t want them to drop out of the market either*.(Housing company).

With today’s socioeconomic systems in Sweden, major retrofit actions to improve housing accessibility were said to risk increased housing costs and force more individuals into homelessness. Although this can be said to be primarily a matter of funding, this nevertheless raised ethical questions about which groups benefited or suffered when accessibility increased in society. Additionally, participants also mentioned to what extent the habits and culture of different ethnic minority groups should be accommodated. Norms for crowdedness were mentioned as an example that could be defined differently across cultures and countries, thus influencing the provision of accessible housing. Another example was the custom of using bidets among certain ethnic groups, while these amenities are not usually installed in current Swedish housing. Here it was noted how housing companies were forced to make a choice about how to allocate space in bathrooms—to take into account people with mobility problems and follow the building regulations, or to take into account the social practices of ethnic minorities. One argument put forward was that while human physiological constitution is an indisputable fact, habits and patterns of life are diverse and changing:
*When we talk accessibility, which we are in favor of, otherwise we would not have been sitting here, you have to keep in mind that there are other stakeholder groups to consider as well; it is not just about disabilities and old age, there are a lot of other things*…(Housing company).
*The difference of course is that there is not much you can do to reverse old age and disabilities. You must base things on those who are most vulnerable… de facto*.(Municipal building administration).

Following this logic, some key actors recommended that in the event of conflicts, priority should be given to the needs of senior citizens and people with functional limitations rather than to individuals and groups who, with their current habits and lifestyles wanted solutions that deviated from the prevailing Swedish standards.

### 5.4. The Aesthetic Dimension

#### Actions with the Intention to Provide Accessible Housing Might Jeopardize Attractiveness

The main topic of the aesthetic dimension was that housing accessibility might jeopardize attractiveness. According to key actors from public housing companies, one aspect in examining the efforts to improve housing accessibility was the possible negative impact on how internal layouts of apartments were provided. While spacious halls and large bathrooms and bedrooms are recommended from an accessibility perspective, the fulfilment of such recommendations bears the risk of reducing the size of the remaining rooms. Consequently, internal layouts could limit the possibility of social gatherings at home and might not be perceived as proportional and attractive:
*For every square meter then, as you said, then you have to narrow down the room there, and then that room becomes slightly smaller, and then suddenly those who are going to rent it think that: No, we probably skip it because that room is too small. They also do not look at this with the bathroom being slightly larger... or if it is good for them maybe in the future. When they look... they focus on the pieces they use today. This is also the problem, that if you are going to have this extended accessibility, then it is the case that you may steal square meters from the other living areas, and then make the rooms not so attractive anymore, because you do not understand really how to get a bed in there*...(Housing company).

As to home care provision, in order not to create negative feelings or stigma, participants discussed that it was important that the dwelling did not give an institutional impression and negative associations with loss of autonomy. Similarly, traces of retrofits or housing adaptations that did not fit the housing context in which they are placed were seen as potentially having a negative impact on the attractiveness of a home and willingness to use supporting housing features, which also may then have additional negative consequences.
*Are the shortcomings repairable or are they irreparable? A home is not the same afterwards, and then... say that when you adapt for an individual, it lasts for maybe five years, then you are dead, and then it must be restored again, because there are no pleasant tracks in general after an* [individual housing] *adaption grant… it’s like a blacksmith’s hammer* [coarsely installed]*, has been there and done it* [installation of grab bars]*, often the devices have an ugly design too. So, it is important to do it* [retrofits] *with a long-term view*.(Senior citizens’ organization representative).

In addition, the key actors argued that attributes such as a private garden or proximity to the sea have a major impact on senior citizens’ decisions to stay in a less accessible villa instead of relocating to a more accessible apartment without such attractive attributes.

### 5.5. The Empathetic Dimension

#### Lack of Empathy Leads to Lack of Understanding of Others’ Situations and Perspectives

The empathetic dimension resonates with discussions about trying to understand and influence other people’s attitudes and feelings towards housing accessibility. The average citizen without any functional limitations as well as many professionals were accused of not thinking proactively enough about housing accessibility. However, there was self-criticism among the participants of the ways in which especially older people were usually approached and informed about such matters. Participants pointed out that having a more sincere and empathetic understanding of other people’s situations would probably improve the chances of achieving a change in attitudes and actions. The ability to put oneself in someone’s place was something that was emphasized as important. For example, understanding the feelings and thoughts that this phenomenon elicited, followed by thoughtful communication, focusing on positive instead of negative aspects:
*When we talk to people aged between 55 and 65 years old, well they can get quite het up, and that is certainly something. There we must think one step further… I think it is a selling point if I can stay in the same place or if I need to move if I know that I can, compared to if I need to stay in the same place because I won’t need so many home care night visits or night camera monitoring, or that I won’t avoid having to go to A&E* [Accident and Emergency] *because of a fall or something. I think that is very good information to tell the elderly about*.(National public authority).

## 6. Discussion

The findings illustrate that multiple critical variables are intertwined in a complex manner within the wicked problem of providing accessible housing for the ageing population. There is no linear cause-effect relation, but a complex web of interdependent variables characterized by diverse as well as more congruent individual thinking, and trade-offs are necessary because there is no optimal solution.

It is noteworthy that the findings from the present study are similar to results presented already 30 years ago [35]—different actors operate in parallel “silos” when it comes to decision-making and realization, and the outset to tackle the problem differs between various perspectives. Such a persistent problem deserves serious attention and action and could benefit from the developments described in “boundary work” [36], which is a concept gaining an increased interest in organization and management science. The concept denotes “*purposeful individual and collective effort to influence the social, symbolic, material, or temporal boundaries; demarcations; and distinctions affecting groups, occupations, and organizations*” [36] (p. 704). Boundary work refers to individual and collective efforts regarding different types of boundaries and, for example, the opening of these. This kind of reasoning is closely related to Brown and colleagues’ [19] framework to address wicked problems and therefore highly relevant to pave the way forward, making use of the findings of our study. A practical way to realize efficient and cost-effective actions to improve accessibility and generate new evidence could be to carry out formative transdisciplinary heuristic evaluations based on construction drawings and visualizations, as early as possible in design processes (see e.g., [37]). 

In line with findings from Imrie [16], our findings reveal that the outset from the perspective of housing companies is that accessibility matters are a biophysical problem that only concerns people with functional limitations, rather than an outset seen from a socioeconomic preventive-health perspective or an ethical and basic human right perspective. Various interest parties and decision-makers affect the provision of accessible housing, and their diverse perspectives should be acknowledged and communicated in closer collaboration to be able to better understand and act upon the wicked problem elucidated by the findings of the present study. 

The biophysical dimension revealed that there is a lack of systematic information about the state of housing accessibility in the ordinary housing stock, as well as a low level of knowledge and insufficient evidence on the health and wellbeing impacts on individuals and society. Here, the application of performance-based assessment methodologies for quality evaluation of built environments (see e.g., [38,39]) might have the potential to contribute to the much-needed evidence base.

The lack of a widely accepted definition of housing accessibility prevails, and there is no consensus about what should be included in measurements of housing accessibility. Relating to such challenges, we emphasize that the absence of reasonable and clear guidelines and goals regarding housing accessibility is in accordance with a previous study elucidating how the definition of standards for housing design influences the prevalence of housing accessibility problems and public health [40]. The opinions of what to include and consider in housing provision policies and practices vary, and housing accessibility relates to many other needs that are important to address. According to Lawrence [41], shared terminology and shared goals that are valid from the micro-scale housing units to the macro-scale national housing stock are a prerequisite for transdisciplinary modes of inquiry, but in reality, that ambition is far from met.

A primary finding is that the largest proportion of content was sorted into the socioeconomic dimension. It is obvious from the categories and quotations used for illustration that in this dimension, there are many critical variables contributing to the wicked problem addressed. With “silo thinking”, distribution of responsibilities and resources, various practices, and competing priorities among actors, the cultural facet of the socioeconomic dimension is obvious. The findings indicating that efforts to improve housing accessibility could reveal or create challenges in other dimensions confirm the wickedness of the problem. In line with Brown and colleagues [19], the findings show that boundaries to knowledge cultures are strong, and that open transdisciplinary modes of inquiry and cross-sectorial communications are needed for mutual learning and development of common understanding of preferred situations, including the economic and social structures. Representatives of all actors and institutions including people facing social, environmental, and health inequalities should be engaged in dialogues to discuss and agree on crucial variables and trade-offs in early planning and provision phases to apply a transdisciplinary approach.

The ethical dimension revealed considerations relating to equality, personal responsibility, responsibility for others as well as fair and reasonable use of resources without compromising people’s freedom of choice. The findings show that decision-making related to housing accessibility involves ethical dilemmas and necessary trade-offs because of the diversity of needs that should be addressed. One such indicated risk was that efforts to improve housing accessibility—based on the principle that everyone, regardless of ability to function, has certain rights [42]—could make it difficult to achieve the principle that all people should be entitled to the right to have a home and the municipalities’ responsibility for all inhabitants [10]. However, the findings in the ethical dimension also revealed creative and responsible solutions to dissolve conflicting principles and address several problems at once, for example motivate “matching” of housing to both resolve accessibility problems for one group of individuals and problems with crowdedness for the other group of individuals. 

The dialogue on matters sorted into the aesthetical dimension concentrated on trade-offs such as negative attitudes to housing features fulfilling accessibility standards but resulting in disproportional interior layouts, stigmatizing housing adaptations and hospital-like dwellings. Current literature highlights the value of aesthetics regarding housing adaptations and the need to involve the users in the decision-making processes to achieve the best possible acceptance of design solutions [20]. 

The empathetic dimension revealed a shared mutual understanding of the challenges inherent in providing accessible housing. Emphasizing the need for empathic communication, the dialogue focused on trying to understand and influence attitudes and feelings about housing accessibility. The findings indicate that empathetic and creative ways to communicate and influence are vital to changing people’s positions, bridging boundaries and striving for preferred situations. In line with Tural and colleagues [20], the findings indicate that communication strategies targeting senior citizens should include positive messages such as independence and health benefits. Once again referring to research on similar matters published long ago [35], while the participants in the present study seemed rather optimistic regarding the possibility to change attitudes, this is a longstanding challenge that has proven very difficult to overcome.

### Strengths and Limitations

The conceptual framework used [19] was instrumental in unwinding and understanding the complexity of providing accessible housing for the ageing population. Our experiences during the process of analysis, as well as the presentation of findings, reveal that the five dimensions are intertwined. As wicked problems are complex issues influenced by a web of interrelated variables that are unpredictable [15], this is not surprising and thus speaks to the relevance of Brown and colleagues’ framework. The use of the five dimensions to address a wicked problem and the deductive approach was helpful in unwinding and visualizing the critical variables for decision-making about the provision of accessible housing.

During the analysis process we identified the socioeconomic dimension as particularly challenging, because, as described by Brown and colleagues [19], it is different from common interpretations and uses. That is, while it does comprise what in the traditional sense is called “socioeconomics”, it also refers to what is usually described as “culture”. That said, during the analysis, it was particularly challenging to distinguish content related to the socioeconomic dimension from content belonging to the biophysical or ethical dimensions. On the one hand, the stringency of our analysis process could be criticized, but on the other hand, this experience has strengthened our impression of the complexity of the critical variables of the wicked problem of providing accessible housing for the ageing population in Sweden.

Further studies to structure and resolve the wicked problem addressed could make use of a multi-criteria analysis approach [43] or an integrated approach on collaborative multi-criteria decision processes [44]. For further studies, such methodologies could potentially be combined with Geographical Information Systems (GIS) [45].

Reflecting further on study strengths and limitations, we perceive the RC method [25] as adequate for dialogue and mutual learning. Considering the purpose of our study, it was a strength that both key actors and researchers took active roles in the dialogue as participants in the RC. The procedure and approach created an open climate with a broad and dynamic dialogue. The dialogue was allowed to be guided by the verbal contributions of all participants, and thus included diverse perspectives and problems. To involve the key actors as much as possible, we did not establish a detailed study protocol in advance. For example, the key actors’ contributions included hosting a session, presenting at sessions, and co-moderating a session. Such a participatory approach can be viewed as both a limitation (i.e., not possible to replicate), a challenge (i.e., make sure that no one takes over too much) and a strength (i.e., reveal the complexity in societal challenges). 

Since the key actors were highly selected, a limitation could be that they may not represent those that potentially could benefit from new knowledge produced. In addition to the fact that the number of participants was low, a noteworthy shortcoming is that we failed to include key actors from the private housing sector, as those invited were not able or willing to participate. Their absence was raised as an issue by key actors from public housing companies with the argument that they are both players in the same market. The findings serve to deepen the understanding of the wicked problem at target, but only represent what was discussed in the context of the present study in Sweden. 

## 7. Conclusions

While discussions regarding the provision of accessible housing for the ageing population among researchers and representatives of housing sector and public institutions are dominated by socioeconomic matters, the multiple dimensions of this wicked problem are intertwined in a complex manner. This is critical for decision-making, which, to a large extent, takes place in parallel organizations and processes with insufficient communication among the actors involved, which speaks to the need for boundary work. Decision-making linked to housing accessibility should not be approached solely considering biophysical or financial variables. Rather, issues related to ethics, aesthetics, and empathy variables that are interrelated should not be ignored. The framework used in this study can serve as a cognitive tool for decision-makers, and the findings could increase the awareness of the diversity of individual thinking involved when addressing this wicked problem. Acting upon the critical variables identified in this study could contribute to progressive decision-making and more efficient ways to develop and provide accessible housing for the ageing population.

## Figures and Tables

**Table 1 ijerph-18-01169-t001:** The definition of “wicked problem” related to the provision of accessible housing.

Definition of Wicked Problem ^1^	Example from the Provision of Accessible Housing
1. There is no definitive formulation of a wicked problem.	The formulation of accessible housing is the problem. The information used to define the problem depends upon a variety of non-academic actors’ and institutions’ idea for solving it.
2. Wicked problems have no stopping rule.	What is considered a “good enough” provision of accessible housing is situational and continuously transforming.
3. Solutions to wicked problems are not true-or-false, but good-or-bad.	Solutions are likely to be differently judged, depending on the special value-sets and ideological predilections of actor and institution groups or their personal interests.
4. There is no immediate and no ultimate test of a solution to a wicked problem.	The full consequences of plans aimed at improving housing accessibility cannot be appraised until the waves of repercussions have completely run out.
5 Every solution to a wicked problem is a “one-shot operation”; because there is no opportunity to learn by trial-and-error, every attempt counts significantly.	Every action taken to improve housing accessibility is consequential for other household members, residents, or visitor. Major retrofit actions to improve housing accessibility affect the residents during the construction process, are expensive, and leave long-standing and irreversible traces.
6. Wicked problems do not have an enumerable (or an exhaustively describable) set of potential solutions, nor is there a well-described set of permissible operations that may be incorporated into the plan.	Many new ideas or efforts to provide accessible housing for the ageing population may become relevant as a resolution. For example, refined grant mechanisms for individual housing adaptations or assistive devices, supports for the housing sector, changes in building regulations or human right laws, or incentives for relocations. However, despite a set of possible candidates the problems could persist. Enlarging the set of solutions as well as choosing which solution to pursue and implement is a matter of judgement.
7. Every wicked problem is essentially unique.	The basis for decision-making related to the provision of accessible housing needs to be local, timely, and put into the context of available resources (e.g., technological, financial) and the specific real-world situation (e.g., social, cultural).
8. Every wicked problem can be considered to be a symptom of another problem.	The level at which problems are settled depend upon the various decision-makers. “Higher-level” problem formulations becomes broad and general, but less operationalizable. On the other hand, solely addressing the problems of providing accessible housing on a too low level and cure symptoms (e.g., individual housing adaptions, assistive devices) can create negative effects on several other variables and making it more difficult to deal with higher level problems.
9. The existence of a discrepancy representing a wicked problem can be explained in numerous ways. The choice of explanation determines the nature of the problem’s resolution.	Diverse worldviews and intentions (e.g., self-interest, profitability, cost-efficiency, health prevention, human rights) are strong determining factors for the various decision-makers. There are no optimal solutions or agreed upon ways to evaluate provision of accessible housing.
10. The planner has no right to be wrong.	Decision-makers become responsible for all the consequences of the actions taken to improve the provision of accessible housing. The increasing pluralism of the contemporary public, who use different and contradicting definitions and scales to assess and judge the consequences of the solutions increases the wickedness of the problem and related dilemmas.

^1^ Rittel and Webber’s ten properties [15].

**Table 2 ijerph-18-01169-t002:** Five-dimensional framework to address wicked problems.

Dimension	Description
Biophysical (measurements)	The biological and physical environment in which an issue is set. Arrived at by observations, measurements, and formal reports.
Socioeconomic (stories)	The social environment, including cultural rules and the socioeconomic systems (a prevailing emphasis in Western culture). Arrived at through reflecting on a cultural framework and/or a personal commitment to a way of life or a religion.
Ethical (principles)	The principles governing relationships between individuals and society and between individuals and the environment.
Artistic/Aesthetic (patterns)	Sensitivity to the patterns in natural and in social systems, arising from the capacity for inspiration within each human being. Arrived at by both expressing and rebelling against cultural norms.
Sympathetic/Empathetic (feelings)	Recognizing a shared understanding with another human being or group. Arrived at through feelings of openness, trust, and shared experience.

Note: After Brown, Lambert & Harris [19] (p. 34–37).

**Table 3 ijerph-18-01169-t003:** Participant characteristics (*N* = 15).

Characteristic	Total (*N* = 15)
Sex	
Men	9
Women	6
Key actors	
Public housing company ^C6, C7, C7^	3
Municipal building administration ^B3, B4^	2
National senior citizens’ organization	1
Municipal health care administration ^B4, C6^	2
Private service provider within the assistive device sector	1
Private architecture and engineering consultancy	1
National public authority (National Board of Health and Welfare)	1
Business developer	1
Researchers	
Lund University employee (authors O.J., M.H. & J.F.)	3

Classification of the municipalities in three main groups (A, B, C) and a total of nine possible subgroups (1–9) on the basis of structural parameters such as population and commuting patterns [30]. ^B3^ Medium-sized towns: Municipalities with a population of at least 50,000 inhabitants with at least 40,000 inhabitants in the largest urban area. ^B4^ Commuting municipalities near medium-sized towns: Municipalities where more than 40% of the working population commute to work in a medium-sized town. ^C6^ Small town: Municipalities with a population of at least 15,000 inhabitants in the largest urban area. ^C7^ Commuting municipalities near small towns: Municipalities where more than 30% of the working population commute to work in a small town/urban area or more than 30% of the employed day population lives in another municipality.

**Table 4 ijerph-18-01169-t004:** Critical variables and categories identified in the material, sorted into the five dimensions of Brown and colleagues’ framework.

Dimension ^1^	Category	Critical Variable
Biophysical (measurements)	Different opinions on the meaning and definition of housing accessibility prevail	Definition of housing accessibility
		Environmental barriers—who is affected?
		Level of detail of housing accessibility
	Systematic inventories are warranted but must be comprehensive	Systematic inventories of environmental barriers
		Degrees of housing accessibility—how to classify and label?
		Embrace variations in human functioning
	Evidence and convincing arguments for housing accessibility are important but lacking	Objective and comparable information on housing accessibility
		Evidence on the benefit to public health and societal economy
Socioeconomic (stories)	The ageing-in-place policy is significant for decision-making	Coherent policy
		Agreed objectives
	Organization and distribution of resources suffer from “silo-thinking”	Transparent cross-sectorial and multilevel communication
		Efficient distribution of resources and responsibilities
		Systems thinking
	Varying practices and competing priorities among the actors	Demand from the market on housing accessibility
		Efficient financial policy
		Awareness, knowledge and competence
	Absence of clear housing accessibility guidelines and goals	Organizational culture
		Cross-boundary collaboration
		Reasonable distribution of responsibilities among the various actors
Ethical (principles)	Balance between individual freedom of choice and societal solidarity	Fair and reasonable use of resources
		Acknowledge individual’s right to chose
	Ambiguous social responsibilities of housing companies	Maintain the welfare state
		The needs and interests of other groups
	Balance between housing accessibility and affordability	Allow for different alternatives
Artistic/Aesthetic (patterns)	Actions with the intention to provide accessible housing might jeopardize attractiveness	The proportions and compositions of rooms
		Affective experiences of housing accessibility—messages sent and received
		Suit the specific context
		Attractive locations and attributes
Sympathetic/Empathetic (feelings)	Lack of empathy leads to lack of understanding of others’ situations and perspectives	Proactive thinking to predict behavior
		Thoughtful communication

^1^ Brown, Lambert & Harris’s five dimensions [19] (p. 34–37).

## Data Availability

The data are not publicly available due to individual privacy, ethical considerations and the fact that data sharing is not in accordance with consent provided by key actors on the use of data.

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
