# Peer review of "Understanding the Wicked Problem of Providing Accessible Housing for the Ageing Population in Sweden"

_ijerph, 2021, doi:10.3390/ijerph18031169_

Round 1

Reviewer 1 Report

The article addresses the original and actual topic of understanding variables for decision-making about accessible housing provision for the ageing population focusing on Sweden. The method of analysis is very accurate, the review suggests few modifications in the introduction and in the last part of the article in order to better highlight the result of the study. The suggestions are related to articles to make them as clear as possible.

In the introduction, I suggest to insert a specific sub-paragraph about Sweden context taking information from ‘Housing Accessibility’, since this topic is also cited in the titled of the article.

In ‘materials and methods’ section, when you cite for the first time the research circle (RC) methodology, I suggest to explain it at least with a sentence, in addition to the reference.

The study reveals that there is a lack of evidence on how housing accessibility impacts on individuals and society’s health and well-being. For this reason, I suggest to clearly state this concept in the discussion section and to propose a practical way to find evidence in further research (e.g. pre and post design assessment of housing accessibility variables to measure the impacts on people’s health). In relation to this, I suggest to consult this article, which addresses the topic of Universal Design evaluation in a practical way, proposing an assessment method for buildings based on major and minor usability problems.

Afacan, Y., Erbug C. (2009) An interdisciplinary heuristic evaluation method for universal building design. Applied Ergonomics 40: 731–744

In the reviewed article emerges that there is a lack of systematic information about the state of housing accessibility, but the use of the five dimensions was helpful in visualizing the critical variables for decision-making about accessible housing provision. However, one of the limitations, as stated in the discussion, might be the stringency of the analysis process. In this regard, for further studies I suggest to consider Multicriteria Analysis approach (Roy,2013; DCLG, 2009), which can be used to structure and solve complex decisions and planning problems involving multiple qualitative and quantitative criteria and the analysis and comparison of the full range of aspects related to a project. This can be done also by involving different stakeholders (Ferretti, 2016).

Roy, B. (2013). Multicriteria methodology for decision aiding (Vol. 12). Springer Science & Business Media.

Department for Communities and Local Government (DCLG) (2009) Multicriteria Analysis: a manual London http://eprints.lse.ac.uk/12761/1/Multi-criteria_Analysis.pdf

Ferretti, V. (2016) From stakeholders analysis to cognitive mapping and Multi-Attribute Value Theory: An integrated approach for policy support, European Journal of Operational Research, Volume 253, Issue 2, Pages 524-541

In relation to this, I suggest to consult the following article which describes how to apply Multicriteria analysis involving different stakeholders for creating a decision support system formed by dimensions, criteria and indicators, similar to dimensions and variables of the reviewed study.

Dell’Ovo et all (2017) Multicriteria decision making for healthcare facilities location with visualization based on FITradeoff method  DOI: 10.1007/978-3-319-57487-5_3

Furthermore, the current study focuses on ‘wicked problems’, however I suggest for further research to consider as variables both negative and positive features of the built environment through a performance based approach. In relation to this, I suggest to consult the two following articles, the first one about a performance-based decision support system for housing renovation, in relation to sustainability and Universal Design, the second one about healthcare facilities for dementia

Kapedani H., Herssens J., Verbeeck G. (2018) The ComfortTool - Assessing perceived indoor environmental comfort improvements in four deep energy home renovations

Brambilla A., Maino R., Mangili S., Capolongo S. (2020) Built Environment and Alzheimer. Quality Evaluation of Territorial Structures for Patients with Dementia. https://doi.org/10.1007/978-3-030-52869-0_15

Finally, the references should be related to the numbers within the article text.

Author Response

Dear Reviewer,

Thank you very much for a fast review process and valuable comments. We have now carefully considered the reviewers' feedback and rectified the shortcomings that have been pointed out to us, and are hereby pleased to re-submit the original research article.

Thank you for your consideration.

Sincerely, on behalf of all the co-authors,

Oskar Jonsson

Reviewer 2 Report

Dear Authors

Thank you for the revisions. Unfortunately, you have missed a few essential points which you must be aware of. In the following, I will take up these points again. Please revise these points before submitting the manuscript again.

Introduction: Missing information about the benefits of this study.

Experimental design: please, try to explain better the procedure. I find it confusing.

Was there inclusion or exclusion criteria in your experiment or study?

Line 211 to 220: How large were effect sizes assumed? Was a sample size calculation done?

Line 230 to 232: Please provide more information about the informed consent.

Limitations must be reported in the discussion section. If the study protocol was not previously, it must be declared as limitation.

Good luck.

Author Response

(The authors gave the same response as above.)

Reviewer 3 Report

This is a well-constructed qualitative study on different types of barriers to the availability of accessible houses to older subjects in Sweden. The clear introduction is supported by relevant references and essential to the comprehension of the framework in which this study was conducted. Results are presented in a clear way and well discussed in the following sections. Conclusions, however, are a bit vague, but it seems inevitable for a wicked problem.

First some general recommendations:

  • The abstract could be improved, providing a more structured summary of results and conclusions
  • Please check reference list, as it does not provide reference number, and it is bothersome to consult.
  • Please revise manuscript for English language, use of repetitions and adverbs mostly. Moreover, in my opinion translation of transcripts could be much improved (feels a little bit strange and sometimes is not fully clear)

Further minor issues to be addressed:

  • Introduction - lines 34-36. The sentence "In addition, poor housing conditions, inequalities and inequity put individuals who belong to one of the above-mentioned groups at risk of ending up in the other groups." sounds a bit rude and might convey a discriminatory message
  • Paragraph 2.1 - lines 85-86: the use of withhold in this sentence is unclear; line 99: please extend CEO acronym
  • Paragraph 3.1 - lines 134-136: this sentence is not so clear when I read it.
  • Paragraph 4.2 - lines 212-213: please specify the "different academic backgrounds" of three researchers; lines 217-219: please specify how key actors benefited from research on housing accessibility (grants? other measures?) if this could be perceived as potentially biasing their role in the RC, this should be discussed in the limitations section, as well as the fact that all key actors were highly selected ("had competence, knowledge, experience of and opinions to issues related to housing and health"), so might not represent the "prototype" of stakeholder for accessible housing.; line 230: please check the number 12, since you included 15 key actors
  • Table 3: please specify which National public authority the key actor was affiliated with
  • Paragraph 4.3 - Please discuss further (here and/or in the limitations section) why academics were both RC active participants and researchers (an independent moderator and/or researcher could be of some use? is this a limitation or a strength of the study?); line 236: JF participated in only one session but the request of participating to all three sessions was not a prerequisite of inclusion in the RC?
  • Paragraph 4.4 - lines 249-250: please provide reference for GDPR.
  • Table 4: please modify the layout since in the present form the table is difficult to read; moreover, Categories and clinical variables provided in the table are extremely vague and should be improved to give the reader a clear "summary" of main findings.
  • Paragraph 5 and all subparagraphs: please uniform transcript presentations (with/without quotes). It could be informative to have at least quoted transcript for all subparagraphs; moreover, the manuscript could benefit from a better balance between the representation of different key actors in the transcripts (the National senior citizen organizations' representative is highly under-represented in my opinion). I also think that quote in lines 384-385 should be longer to really refer to "counter-productive construction projects".
  • Paragraph 5.5: i find it lacking a little bit of depth and worth a longer discussion.
  • Reference 1 - line 713 please provide relevant link: https://www.who.int/publications/i/item/9789241550376
  • References: for all links please update date last accessed (I personally checked and are all accessible as of today, Jan 5th 2021.

Author Response

(The authors gave the same response as above.)
